# Neural traces of composite tasks in complex task representation in the human brain reflects learning performance

**Woo-Tek Lee**[1,2,3], **Eliot Hazeltine**[1,2,4], **Jiefeng Jiang**[1,2,4*]

1 Department of Psychological and Brain Sciences, University of Iowa, Iowa City, Iowa, United States of America, 2 Cognitive Control Collaborative, University of Iowa, Iowa City, Iowa, United States of America, 3 Behavioral-biomedical Interface Training Program, University of Iowa, Iowa City, Iowa, United States of America, 4 Iowa Neuroscience Institute, University of Iowa, Iowa City, Iowa, United States of America

* jiefeng-jiang@uiowa.edu

## Abstract

Task knowledge can be encoded hierarchically such that complex tasks can be built by associating simpler tasks. This associative organization supports generalization to facilitate learning of related but novel complex tasks. To study how the brain implements generalization in hierarchical task learning, we trained human participants on two complex tasks that shared a simple task and tested them on novel complex tasks whose association could be inferred via the shared simple task. Behaviorally, we observed faster learning of the novel complex tasks than control tasks. Using electroencephalogram (EEG) data, we decoded constituent simple tasks when performing a complex task (i.e., EEG association effect). Crucially, the shared simple task, although not part of the novel complex task, could be reliably decoded from the novel complex task. This decoding strength was correlated with the EEG association effect and the behavioral generalization effect. The findings demonstrate how task learning can be accelerated by associative inference.

## Introduction

Humans have the remarkable ability to efficiently learn new complex and cognitively demanding tasks. To do this, we organize task representations hierarchically [1–4], such that complex tasks can be represented by associating simpler ones. That is, new complex tasks can be learned efficiently by associating existing simple task representations and inferring their direct and indirect connections [5,6]. For example, a barista can learn how to make a 'café latte' by associating the existing skills of making espresso and steaming milk. This ability is analogous to studies of human memory, where researchers have found that associating multiple memory items enables us to adaptively adjust behavior by predicting future events [7] or inferring novel information [8]. This associative feature of memory representations has been extended

**Data availability statement:** All data and analysis codes related to this paper are available and archived in Open Science Framework at (https://doi.org/10.17605/OSF.IO/MZF4A).

**Funding:** This project was supported by the National Institute of Mental Health (R01MH131559 to J.J.). The funder had no role in study design, data collection and analysis, decision to publish, or preparation of the manuscript.

**Competing interests:** The authors have declared that no competing interests exist.

**Abbreviations:** EEG, electroencephalogram; ICA, independent component analysis; lPFC, lateral prefrontal cortex; MTL, medial temporal lobe; PFC, prefrontal cortex; RSA, representational similarity analysis; SEM, standard error of means; VIF, variance inflation factor.

to incorporate the cognitive control parameters and task demands [5,6,9–13] that support adaptive, goal-directed behavior. Although the lateral prefrontal cortex (lPFC) has been shown to be involved in representing task rules for goal-directed behavior [14], little work has addressed how simple tasks can be associated and generalized for building of complex task representations.

This associative feature of task representations allows complex tasks to share components, which enables humans to generalize from previous experience and quickly adapt to new, related tasks [15]. For example, humans can extract structure from experiences and infer novel knowledge from the structure, a phenomenon sometimes also termed zero-shot learning [16–18]. While generalization can occur via different mechanisms, such as rule-based or similarity-based [19], generalization through association requires additional inferential processes as it allows for learning without direct association or experience. This generalization in task learning may be attributed to associative inference, which refers to the ability to infer novel, indirect associations from partially overlapping associations [9,11,20–27]. For example, the barista who is skilled at the 'café latte' task (espresso + milk) and the 'café mocha' task (espresso + chocolate) can infer how to make a hot chocolate via the 'making espresso' task shared by the 'café latte' and 'café mocha' tasks. In the laboratory, we reported that generalization via associative inference accelerates task learning and improves task performance [6]. Although generalization by associative inference plays an important role in task learning [3,6,9], how it is implemented in the brain is understudied.

Two prominent theories of association and generalization in episodic memory offer insights into the processes supporting abstract task learning and generalization. Integrative encoding theory [22] posits that repeated co-activation of items (e.g., A and C) will create an associated memory representation (e.g., AC). Moreover, integration can occur between associated item pairs that share an item (e.g., AB and BC), which leads to an integrated representation (e.g., ABC) that encompasses all three memory items [20,22,25]. The integrated representation will be retrieved if a subset of the items is present. For example, presentation of AC is expected to reinstate item B as well as items A and C. On the other hand, recurrent interaction theory [21] proposes that an association between two memory items forms a conjunctive representation, and these conjunctive representations are connected to each of the constituent memory items. If a representation is activated (e.g., by sensory input), its connected representations will also be activated (though to a lesser extent). For example, if two conjunctions AB and BC share a memory item B, activating the conjunction AB will also activate memory items A and B. B will further activate conjunction BC, which in turn activates C. In the end, the co-activation of A and C will facilitate the formation of an indirect association of AC.

While these two major theories differ in terms of their proposed mechanisms of generalization, they share an important prediction about how the generalization occurs between associated memory representations via associative inference. When there are multiple associated item pairs that share a component (e.g., A-B and B-C association), activation of nonoverlapping items (i.e., A and C) should reinstate the

latent (i.e., not presented; B) item, because of the pattern completion processes of integrated representation (integrative encoding; [22]), or feedback signals from conjunctive representations that are linked with activated items (recurrent interaction; [21]).

Applying this idea to the context of task learning, we predict that the reinstatement of the latent task from the complex task is expected to be key for generalization, which relies on the associations between simple task representations. Moreover, both theories predict that stronger latent task reactivation will lead to more efficient learning, as the reactivation supports the inference generated by indirect associations between constituent task representations via stronger activation of integrated representations [20,22,25] or stronger reinstatement of constituent representations that are connected with latent representation [21]. However, while the reinstatement of memory items for generalization has been extensively studied, whether the same neural mechanism applies to *abstract task representations*, and if so, how the latent representations support efficient task learning, has not been directly tested. Here, we predict that the co-activation of simple task representations during complex task learning leads to associations between simple tasks and that these associations can facilitate the new complex learning consisting of indirectly associated simple tasks. Based on these predictions, we hypothesize that the generalization will lead to reinstatement of latent task representation. Additionally, we hypothesize that stronger neural evidence of associative activation of latent task will predict stronger generalization.

To test these predictions, we leveraged a behavioral paradigm of generalization in task learning [6]. In this paradigm, subjects performed tasks that require to attend task-relevant feature(s) based on the cue(s). While the 'task' in this paradigm is strongly intertwined with feature-based attention, our use of the term is consistent with classic task-switching literature [28–30], where a 'task' requires the process that enables the efficient guidance of attention toward task-relevant feature. Furthermore, the task-switching literature provides convincing evidence that a 'task' cannot be fully explained by feature-based attention [29–32]. For instance, mixing costs, which refer to the cognitive cost observed when performing a task in a mixed-task block (i.e., on each trial one of multiple possible tasks is performed) compared to a single-task block, cannot be fully explained by shift of feature-based attention, since the cost occurs even when participants repeat the same task, implying no change in feature-based attention [29,30].

During performance, we obtained high temporal resolution and high-density electroencephalogram (EEG) data, to which we applied decoding and representational similarity analysis (RSA, [10, 33]). To preview the results, the behavioral data revealed faster learning of complex tasks consisting of indirectly associated simple tasks compared to control tasks consisting of equally practiced simple tasks that were not indirectly associated. This result replicates the behavioral generalization effect reported in Lee, Hazeltine and Jiang [6] and suggests that participants learned feature-based tasks efficiently by optimizing cognitive control mechanisms. We then observed robust decoding of constituent simple task representations from the complex task EEG data, which validates that our analysis is sufficient to capture dynamics of abstract task representations. Critically, we can decode the latent task (e.g., B) that is reinstated by the indirect association in the complex task (e.g., AC) EEG data. We further found that the strength of the latent task representation in complex task EEG data is linked to both the strength of the constituent simple task representations in complex task EEG data and generalization performance. In sum, the results suggest that complex task learning results in a network representing associated information that enables generalization via the reinstatement of the latent task.

## Results

### Experimental design

Participants ($n = 40$) first learned six simple tasks (coded as A-F), each of which was a feature-based attention task defined on one out of the six stimulus features (Fig 1A). On each trial, participants judged whether a cued feature appeared in a forthcoming target stimulus (Fig 1B). After the simple task phase, participants learned complex tasks, each consisting of two simple tasks. On each trial of complex task, participants followed the 'exclusive or' design, judging whether or not only one of two cued features matched the target stimulus. Because there were two possible responses,

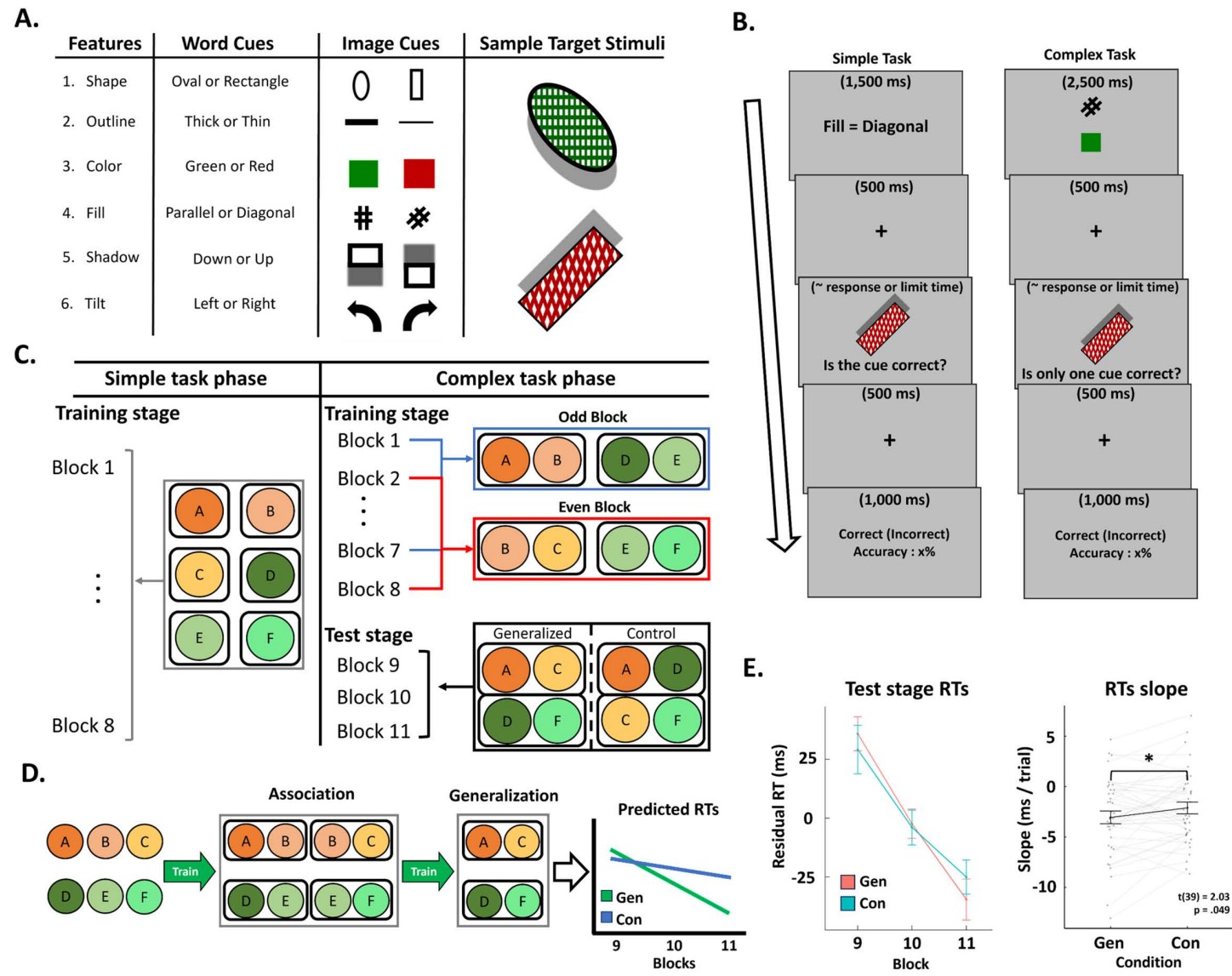

**Fig 1. Experimental design and behavioral results. (A)** Cue and target stimuli. Each cue indicated one stimulus feature and was presented as either text or image. Two example stimuli are illustrated in the rightmost column. **(B)** Trial events of the simple and complex tasks. In the simple task, participants first encountered a single cue that predicts a feature of an upcoming stimulus. After a delay of 500 ms, the target stimulus was presented until response or a deadline. Participants judged if the cue matched the target with button press. Visual feedback was provided after another 500 ms delay. The complex task is identical to the simple task except for two changes. First, participants encountered two cues instead of one. Second, participants decided whether only one cue matched the target (i.e., the 'exclusive or' rule). **(C)** Illustration of the experimental design. **(D)** Predicted dynamics of task representations in the complex task phase. Participants were expected to associate simple task representations to build a complex task representation and to further generalize two complex tasks via associative inference. Thus, we predicted that the generalizable complex tasks present faster decrements in RT than the control complex tasks. **(E)** Block-wise RTs (left, group mean and standard error of means, SEM) and slope of RTs (right, group mean and SEM, dots represent individual slopes) of complex tasks for each condition during the test stage. The data and code needed to generate this figure can be found at https://doi.org/10.17605/OSF.IO/MZF4A.

the chance level of correct response was 50%. Within each feature, both feature values were presented equally frequently during the complex task phase. During the training stage of the complex task phase, different complex tasks, denoted as the pair of constituent simple tasks (e.g., AB means a complex task consisting of simple tasks A and B), were presented

on alternate blocks (odd blocks: AB, DE, even blocks: BC, EF; Fig 1C). In the training stage, there were two sets of complex tasks, such that each set contained two complex tasks sharing one simple task (e.g., set 1: AB and BC, set 2: DE and EF). After the training stage, participants performed four new complex tasks (AC, DF, AD, and CF) in the test stage (Fig 1C). Two of the new complex tasks could be generalized via associative inference because both simple tasks were associated with the same simple task in practiced complex tasks (e.g., AC from AB and BC; Fig 1D) and are hence termed generalizable tasks. In contrast, the other two new complex tasks (e.g., AD and CF) could not be generalized using associative inference, as the constituent simple tasks did not overlap during the training stage. These were termed control tasks. Generalizable and control complex tasks were presented equally frequently within each block in random order. Within this behavioral framework, we defined task representation as cognitive processes involving task-relevant feature information and further posited that a major component of learning involves the improved application of cognitive control and associative mechanisms, leading to optimized feature-based selection and processing and enhanced behavioral performance, based on previous research [6]. This design enabled us to investigate the behavioral generalization effect by comparing performance between generalizable and control tasks, while also examining the temporal dynamics of associative and generalization effects through EEG data. We note that the assignment of each simple task to a specific label (e.g., A, B, C, D, E, or F) was random for each participant. For additional confirmation, we visualized frequencies of each simple task assigned to each specific label (S6 Fig). This histogram showed no systematic bias of simple task assignments. This result supports that the reported findings are not mainly driven by a particular simple task.

## Behavioral results

Group-average accuracy for both simple and complex tasks stayed above 80% throughout the experiment (S1 and S2 Figs). Performance varied significantly across individual simple tasks, particularly for the 'fill' task (S1 Table), so we conducted additional analyses to address potential confounding effects that the EEG results were driven by these behavioral differences, as detailed in the 'Above-Chance Decoding in Simple Task EEG Data' section.

To assess the generalization effect, we compared the performance of the generalizable and control conditions in the test stage (Table 1). Accuracy for the generalization and the control conditions did not differ significantly ($t(39) = 1.56$, $p = .126$, Cohen's $d = .25$). For the RT analysis, we followed our previous procedure [6] and regressed out confounding factors such as performance differences among simple tasks, response switch costs and so on (see Behavioral analysis in Materials and methods for details). To control for potential high multi-collinearity between confound regressors, we calculated variance inflation factor (VIF) scores for each regressor and confirmed that all VIF scores are lower than 2 (S2 Table). The resulting residual RTs were used to test the hypothesis of generalization. Replicating our previous finding [6], we found significantly faster decrease in RT over time ($t(39) = -2.03$, $p = .049$, Cohen's $d = .32$) in the generalizable

**Table 1. Behavioral results of the test stage in complex task phase. Group mean (SEM) of accuracy and residual RT in each condition (Gen = generalizable, Con = control) and block of the test stage.**

| Condition | Block | Residual RTs (ms) | Accuracy (%) |
|---|---|---|---|
| **Gen** | 9 | 35.76 (7.20) | 78.54 (7.69) |
| | 10 | −2.17 (6.27) | 79.17 (7.92) |
| | 11 | −34.53 (8.68) | 81.77 (7.07) |
| | Slope | −3.05 (0.62) per trial | 1.67 (1.11) per block |
| **Con** | 9 | 28.75 (9.91) | 74.69 (7.19) |
| | 10 | −4.23 (7.77) | 79.48 (5.80) |
| | 11 | −24.19 (7.36) | 79.17 (7.55) |
| | Slope | −2.11 (0.58) per trial | 1.88 (1.11) per block |

complex tasks than in the control complex tasks (Fig 1E), indicating faster learning for generalizable than control complex tasks. To account for the nested design of the present experiment, we performed a linear mixed-effect model analysis and found a consistent significant effect of learning speed ($t(4398.14) = -2.10$, $p = .036$).

## Decoding simple task EEG data

For each participant and each time point (i.e., a 50 ms bin) of the simple task phase (Fig 1C), we trained a multiclass decoder on the EEG data to classify the six simple tasks. The decoding analysis was performed after the onset of the target stimulus (Fig 1B), so the decoder performance was not affected by perceptual processes related to the cue. Cross-validation decoding accuracy of simple tasks increased shortly after the target onset and was significantly above-chance after 50 ms, peaking at around 300 ms (Fig 2A, left panel). The longest epoch that showed significant decoding performance (indicated in red line in left panel of Fig 2A) survived nonparametric permutation test for multiple-comparison correction ($p_{corrected} < .001$). The topographical map of the averaged pattern matrix [34] shows strong absolute weights in occipital and posterior electrodes, with notable contributions from left frontal electrodes, suggesting both sensory and cognitive processes (Fig 2A, right panel). This result suggests that the simple task representations can be robustly decoded in EEG data after target stimulus onset during their performance.

Participants were required to perform the simple tasks with the same response deadline (see Materials and methods), which likely contributed to the significantly different behavioral accuracies between simple tasks (S1 Table). To address the potential confounding effect in the decoder's performance caused by the different number of trials across simple tasks, we calculated decoding accuracies separately for each simple task. Although the decoding accuracies differed among simple tasks, all tasks showed significantly higher decoding accuracies at most time points. More importantly, the decoding accuracies between simple tasks did not correlate with their behavioral accuracies, and the least accurately performed 'fill' task showed similar, or higher, decoding accuracies compared to other simple tasks (S3A Fig). The temporal generalization analysis in each task further showed no systematic difference in their temporal dynamics of decoding performance across different simple tasks (S4 Fig). Overall, this result indicates that the observed difference in behavioral performance between simple tasks did not systematically confound the decoders' performance.

## Overview of cross-phase EEG decoding analysis

To assess the reinstatement of simple tasks during the performance of complex tasks, we trained decoders on simple task phase EEG data, as described above, then tested on complex task phase data (see Materials and methods), only using correct trials in both simple tasks and complex tasks. The mean (SD) number of trials included in the analysis was 293.48 (19.15) for the simple task phase, 215.05 (20.10) for the complex task training stage, and 52.68 (5.56) for the complex task test stage.

To test the association effect (e.g., decoding simple tasks A and C from complex task AC), the decoders were tested on complex task EEG data from the training stage (blocks 1–8). To test the generalization effect (e.g., decoding simple task B from complex task AC), the decoders were tested on the test stage of the complex task phase (blocks 9–11, Fig 1C). For each test time point, the decoder produced a probabilistic prediction of each simple task being represented, which sums up to 1 across all simple tasks in each trial. We logit-transformed the decoding probability data to prevent issue caused by the fact that the sum of all classes' probability is 1. Next, we regressed the logits against a linear model with predictors representing association effect (i.e., constituent simple task), generalization effect (i.e., the latent simple task), and the baseline of the nonassociated latent task (e.g., E in AC complex tasks; see Representational similarity analysis in Materials and methods for more details). This baseline was included to control for any potential confounding effect in the generalization effect in the test stage due to the higher presentation frequency of the latent tasks (i.e., B in AC and E in DF) than other simple tasks during the training stage (Fig 1C) because the presentation frequency may affect the decodability of simple tasks.

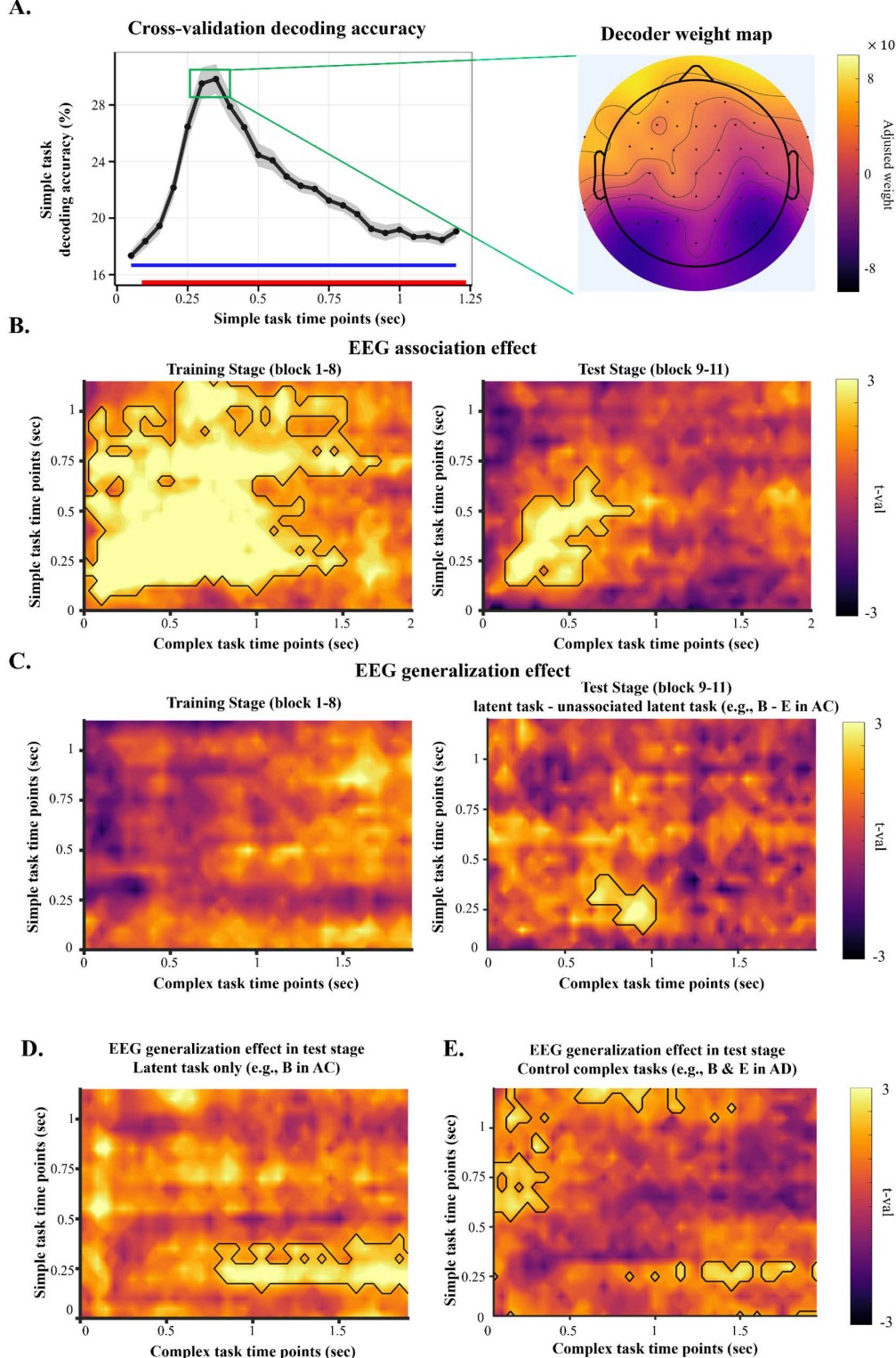

**Fig 2. EEG decoding results. (A)** Simple task EEG data decoding results. Left: decoding accuracy over time within a trial. The gray shade indicates SEM of the decoding accuracy. The blue line indicates chance-level decoding accuracy. The red line indicates time points showing significantly above-chance decoding accuracy. Right: visualization of mean electrode-wise contributions to the decoders based on pattern matrix [34]. **(B)** and **(C)** EEG

association and generalization effects of complex task EEG data. In each panel, decoding performance (averaged *t*-values from trial-level decoding analysis, see Materials and methods) is plotted as a function of the time point of training (Y-axis) and the time point of test (X-axis). (B) T-values indicating decoding strength of constituent simple task representations from complex tasks in the training stage (left panel; e.g., A and B in AB) or in the test stage (right panel; e.g., A and C from AC). **(C)** *T*-values indicating decoding strength of the latent simple task in the training stage (left panel; e.g., C in AB), or in the test stage compared to nonassociated latent simple task (right panel; e.g., B–E in AC). The time points are time-locked to the stimulus onset. **(D)** *T*-values indicating decoding strength of post-hoc decoding analysis testing above-chance decoding of the latent task (e.g., B in AC). **(E)** Results of decoding the latent tasks in control complex task (e.g., B and E in AD). The data and code needed to generate this figure can be found at https://doi. org/10.17605/OSF.IO/MZF4A.

The trial-wise regression coefficient of each predictor was averaged for each participant and tested against 0 using a one sample *t*-test. The presence of an association effect and a generalization effect in training stage predicts that beta values will be greater than 0. For the generalization effect in the test stage, we subtracted the effect of the baseline regressor, thus testing the prediction that decoding performance of the associated latent task will be greater than that of the nonassociated latent task (e.g., B minus E in AC complex tasks). We used the *t*-values to quantify the strength of the association and generalization effects. For the multiple-comparison corrections, we applied a nonparametric permutation test [35].

## EEG association effect

We first investigated the activation of constituent simple task representations to test the hypothesis that constituent simple task representations would be decodable during complex task trials. The purpose of this analysis was to assess the association between simple task representations for complex task learning from EEG data and to validate the simple task decoder's performance when applied to the complex task EEG data. The decoding strength of constituent simple tasks can relate to the associations between task representations in two ways. One possibility is the subsequent memory effect [36–40]. In this study, the subsequent memory effect posits that EEG decodability reflects neural representation strength of the constituent tasks, which in turn is linked to the encoding strength of the association between constituent tasks. Alternatively, performing constituent tasks to complete the complex task may form associations between the constituent tasks. The association, when pair with one constituent task as the cue, may reinstate the other constituent task, leading to stronger task representation on top of that caused by executing the other constituent task.

Fig 2B shows the time points that the constituent simple task representations can be decoded from the complex task EEG data (e.g., A and B from AB) in the training and test stages, respectively. In the training stage, constituent simple task representations were decoded above-chance level from complex task EEG data between 150 ms and around 1,200 ms post target onset when the decoders were trained between around 150 ms and 1,000 ms post target onset ($p_{corrected} < .001$; Fig 2B, left panel). In the test stage, this EEG association effect remained significant ($p_{corrected} = .041$; Fig 2B, right panel) yet was reduced, significant between around 150 ms and 700 ms post target onset in training data and between 150 and 550 ms post target onset in test data. These results indicate that complex task execution involves constituent simple task activation, which is more extensive in the training than in the test stage.

## EEG generalization effect

While representing complex tasks as associations between simple task representations can be beneficial for their initial learning, generalization can support the learning of a novel complex task if its associations can be inferred from other learned complex tasks. Both integrative encoding theory [22] and recurrent interaction theory [21] predict the reinstatement of the latent task representation during test stage complex task trials (e.g., B from AC), despite the fact that the latent task is not present on the trial. To test this, we applied decoding analysis and RSA to assess the reinstatement of the latent simple task representations in the test stage complex task EEG data.

Consistent with the prediction, latent task (e.g., B in AC) representations, compared to nonassociated latent task (e.g., E in AC), were better decoded between 850 ms to 1,000 ms post target onset in the test stage using decoders trained between 200 ms and 300 ms post target onset (Fig 2C, right panel, $p_{corrected}$ = .048). While both theories predict that latent task representation should be presented in the test stage, integrative encoding theory further posits that integrative memory can form when participants experience both complex tasks that share a simple task [11,22]; e.g., AB and BC) during the training stage. This suggests that generalization may occur during the training stage, starting from the third block. To test this prediction that generalization occurred during the training stage, we tested the decoding of the latent task in the training stage (e.g., C from AB). In contrast to the test stage data, no time points reached statistical significance after the cluster-based nonparametric permutation test.

As a confirmatory analysis, and to investigate temporal dynamics of associated latent task reactivation (e.g., B in AC) compared to unassociated latent task reactivation (e.g., E in AC), we conducted a post-hoc analysis, assessing latent task reactivation without the baseline regressor for the nonassociated latent task (Fig 2D). In this post-hoc analysis, latent task representations were decoded above-chance level between 850 ms and 2,000 ms post target onset in the test stage using decoders trained between 200 ms and 300 ms post target onset (Fig 2, right panel, $p_{corrected}$ = .047). This result supports the main EEG generalization effect by showing overlapping time points between 850 ms and 1,000 ms post target onset in the test stage. Furthermore, this result suggests that the latent task reactivation in later time points in complex task EEG data (1,000–2,000 ms) might be confounded by the higher presentation frequency of latent task as supported by the above-chance decoding of the unassociated latent task after 1,000 ms of target onset (S6 Fig). Specifically, latent task in test stage presented twice more often than the other tasks during the training stage, which might evoke late sustained activation during the test stage, regardless of whether the latent task that is relevant to the current complex task [41,42].

As the control complex task also contained simple tasks with learned associations (e.g., A-B and D-E in AD complex task), it is possible that these latent tasks (B and E in AD) were reactivated by these associations. To test this, we regressed the logits of decoding performance for each task against a linear model with regressors for association and generalization effects in control complex tasks, and tested decoding performance of the latent simple tasks in the test stage complex task EEG data (Fig 2E). None of the clusters survived after controlling for multiple comparisons ($p_{corrected}$ = .190). Additionally, we analyzed latent task reactivation by averaging its activity within the time window that showed significant EEG generalization effect in our main analysis (Fig 2C, right panel). A t-test performed on these values revealed no significant effect (t(39) = 1.63, p = .110). However, the difference between latent task reactivation in control complex task and generalizable complex task did not show significant effect (t(39) = 0.57, p = .571). The results did not support the reactivation of latent tasks in control complex tasks.

## Correlation between EEG association and generalization effects

Two major accounts of generalization in episodic memory [21,22] hold that the association effect (i.e., decodability of constituent simple tasks during complex task performance) lays the foundation of generalization effect (i.e., decodability of latent task from complex task in the test stage). Therefore, we predicted that the magnitude of the association effect should correlate with magnitude of the generalization effect. To test this correlation, we first calculated averaged EEG generalization effect for each participant within a time window defined in the following manner (Fig 3A): Based on the EEG generalization effect in Fig 2C and 2E, we calculated the mean EEG generalization effect within the time window that was significant in both main and post-hoc EEG generalization effect: 200–300 ms post target onset for the simple task data and 850–1,000 ms for the complex task data. The decision to use 200–300 ms post target onset for the simple task EEG data is further supported by simple task decoding analysis, which showed the most robust decoding performance in that time window (Figs 2A and S4). For consistency, the EEG association effect was trained on simple task EEG data from the same window (200–300 ms post target onset) and tested on time points showing statistically significant association effect, which starts at 100 ms (Fig 2B, left). The test window was truncated at 850 ms to avoid temporal overlap between the periods used to measure the association and generalization effects.

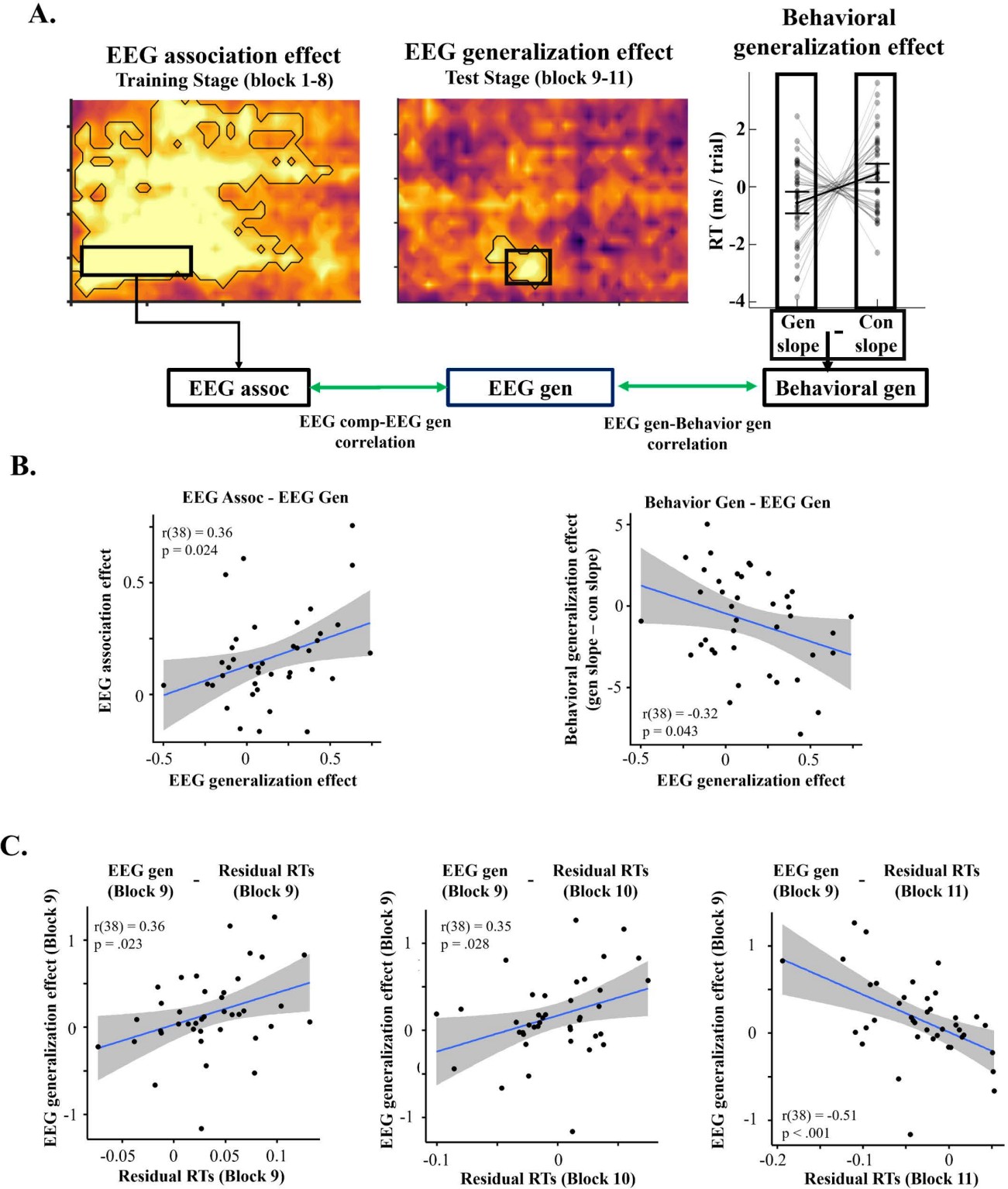

**Fig 3. Correlation analyses results. (A)** Data used for correlation analyses. For each individual, EEG association effect (left panel), EEG general-ization effect (center panel), and behavioral generalization effect (right panel) were used. The EEG association effect was calculated by averaging single-trial *t*-values from the decoders trained on simple task time points between 200 ms and 300 ms and tested on training phase complex task time points between 200 ms and 1,000 ms. The EEG generalization effect was calculated by subject-mean single-trial *t*-values from the decoders to be trained

on simple task time points between 200 ms and 300 ms and tested on test phase complex task time points between 850 ms and 1,000 ms. The behavioral generalization effect was calculated as the difference in RT slope (Fig 1E) between generalizable and control conditions. **(B)** Scatter plots showing individual EEG association and EEG generalization effects (left) or EEG generalization effect and behavioral generalization effects (right). **(C)** Scatter plots showing individual EEG generalization effect in block 9 and residual RTs for blocks 9, 10, and 11, presented from left to right. The data and code needed to generate this figure can be found at https://doi.org/10.17605/OSF.IO/MZF4A.

Consistent with the prediction, EEG association effect in the training stage showed a significant positive correlation with the EEG generalization effect ($r(38) = 0.36$, $p = .024$; Fig 3B, left panel). Overall, these results indicate that the association of simple task representations during the training stage of the complex task phase was linked to the EEG generalization effect in the test stage.

### Correlation between behavioral and neural effects of generalization

We expected that the EEG generalization effect would benefit performance of generalizable complex tasks, whether it is through stronger activation of the integrated memory representation with pattern completion process (integrative encoding) or through stronger activation of the conjunctive representation that is connected with latent memory representation [21]. Therefore, we conducted an exploratory analysis to investigate the behavioral relevance of the EEG generalization effect. We first assessed individual behavioral generalization effects using the RT slope difference between generalizable and control conditions (generalizable − control) and correlated this behavioral generalization effect with the EEG generalization effect from the previous analysis (Fig 1E). The slope differences showed a significant negative correlation with the EEG generalization effect ($r(38) = -0.32$, $p = .043$), indicating that a stronger EEG generalization effect is linked to larger difference in RT decrease in generalizable tasks compared to control tasks (Fig 3B, right panel). The accuracy difference between generalizable and control complex tasks did not show any significant effect with EEG generalization effect ($r(38) = 0.11$, $p = .506$). This result indicates that reactivation of latent task in test stage supports the overall learning speed of generalizable complex tasks.

### Early interference and late facilitation from latent task reactivation

The results above suggest a link between the reactivation strength of the latent task in the brain and the behavioral generalization effect in task learning. However, it remains unclear why the reinstatement of latent task representation, which is not required for the current complex task, benefits performance of generalizable complex tasks. One might hypothesize that the reinstatement of the unneeded latent task would interfere with complex task performance. To investigate potential interference and generalization effects and their relation to latent task reactivation, we conducted an additional cross-subject correlation analysis, using mean residual RT in generalizable complex tasks and EEG generalization effect in each block and tested whether the EEG generalization effect in earlier blocks covaries with residual RTs of the current or later blocks. We hypothesize that latent task reactivation in one block will be correlated with slower or faster RTs in another block as a result of interference or generalization effect, respectively.

Only the EEG generalization effect in first block showed significant correlation with residual RTs in each block, but the direction of the correlation varied with block. Specifically, we found a significant positive correlation between EEG generalization effect in first block and residual RTs in first block ($r(38) = 0.36$, $p = .023$) and second block ($r(38) = 0.35$, $p = .028$), consistent with an interference account that stronger reactivation of the latent task is linked to slower responses (Fig 3C, left and middle panel). However, EEG generalization effect in block 1 showed a significant negative correlation with residual RTs in block 3 ($r(38) = -0.51$, $p < .001$), which suggests that participants with stronger latent task reactivation in block 1 showed faster residual RTs in block 3 (Fig 3C, right panel). EEG generalization effect in other blocks failed to predict residual RTs. Overall, these results suggest that latent task activation evoke interference in short time scale but benefits their behavioral by supporting generalization in later stage.

## Discussion

We investigated the role of task representation reinstatement in supporting association and generalization in hierarchical task learning. To this end, we employed an established experimental paradigm that involved six simple tasks that could be combined to form complex tasks. Participants were first trained with simple tasks, then learned complex tasks that consisted of two simple tasks. Each complex task shared a simple task with another complex task (e.g., AB and BC) in the training stage. In the subsequent test stage, participants performed generalizable complex tasks that consisted of indirectly associated simple tasks (e.g., AC) and control complex tasks that consisted of nonassociated simple tasks (e.g., AD). Note that the generalizable complex tasks and control complex tasks are composed of constituent simple tasks that have received equivalent amounts of practice. Nonetheless, we hypothesized that participants would learn novel complex tasks efficiently by (1) associating learned simple task representations and (2) generalizing complex tasks via associative inference. Behaviorally, we successfully replicated the behavioral finding of greater slopes of decreases in RT for generalizable complex tasks than control complex tasks [6], thus supporting generalization and association (as association is the foundation of generalization).

### Potential neurocognitive mechanisms of generalization

From the EEG data, simple task representations were reliably decoded from the complex task EEG data, supporting the association of simple tasks during both the training and test stage. These findings also validated our approach of applying simple task decoders to complex task EEG data.

Critical for linking these data to generalization phenomena, we identified a significant latent task representation during the test stage but not during the training stage in complex task EEG data. We also observed that EEG generalization effect in test stage is positively correlated with EEG association effect in training stage, and negatively correlated with learning slope of residual RTs in test stage (Fig 3B). Notably, this latent task was not required for new complex tasks, yet its reinstatement during the test stage supports the generalization of complex task representations (Fig 3B). Regarding how the reinstatement of the latent task facilitates generalization, the big-loop recurrence theory [21,43,44] provides a possible account. The theory suggests that the process of memory retrieval in the MTL does not end after the associated item is retrieved, but recurs using the retrieved item as additional cue. In our experiment, the reinstated latent task may contribute to generalization by becoming the cue for the future cued retrieval through the big-loop mechanism in the MTL. In turn, the constituent simple tasks will be reinstated in the next retrieval, adding to their representation strength and subsequently benefiting the learning of the generalizable complex tasks, a process consistent with the recurrent interaction theory. Alternatively, the reinstated latent task may facilitate integrative encoding involving the two constituent tasks, thus accelerating generalization by strengthening the association between constituent tasks, a process that supports the integrative encoding theory.

A key remaining question is how to distinguish between integrative encoding theory and recurrent interaction theory within our complex task learning paradigm. One possible way to dissociate the two theories is by examining the timing of the reactivation of the latent task. The integrative encoding theory predicts that the latent task activation arises as early as the training stage, as the partially overlapping associations (e.g., A-B and B-C) are integrated into a conjunctive representation. On the other hand, the recurrent interaction theory posits that latent task reactivation occurs only at the test phase, when the generalizable complex task can be inferred. We did not find any significant clusters of latent task reactivation during the training stage, which is in line with the recurrent interaction theory. However, the null finding is not conclusive. Future studies designed to directly answer this question are needed to address latent task reactivation in the training phase.

Another approach may be to examine the specific brain regions involved in generalization during task learning. A recent study [45] reported a functional gradient within the hippocampus, with posterior regions supporting smaller-scale associations (e.g., A-B, B-C) and the more anterior hippocampus region supporting more integrative

representations (e.g., A-B-C) in narrative association task. Future neuroimaging studies with higher spatial resolution (e.g., those using fMRI) could provide more evidence that can distinguish two theories. Specifically, such studies could test whether the generalizable complex task learning relies on different subregions of hippocampus. For instance, involvement of the posterior hippocampal regions may suggest that generalization relies on local, small-scale representations, thus selectively supporting the recurrent interaction theory. On the other hand, engagement of the anterior regions may indicate that generalization is attributable to integrative representations, thus favoring the integrative encoding theory.

In addition to the correlation between EEG generalization effect and learning slope of residual RTs, we observed a significant correlation between the EEG association and generalization effect, suggesting that association may support generalization. Specifically, the EEG generalization effect in the test stage correlated with the EEG association effect in the training stage, such that a stronger association of simple tasks during the training stage leads to stronger activation of the latent task representation during the test stage. Underlying the connection between the association and generalization effects are the associations linking the constituent simple tasks. In particular, when a simple task is shared by multiple complex tasks, it bridges multiple associations, which then form a network [21]. A remaining question is how to efficiently organize this network. One possible approach is to embed the network into a cognitive map, implying that spatial and nonspatial information can be organized into a map-like space for goal-directed behavior [46–50]. This would allow for efficient inference [47,51]. Another nonexclusive approach is representation compression [52], which refers to increased similarity between neural representations of associated items (tasks in the context of the present study), leading to facilitation of generalization.

## Generalization and interference effects due to latent task reactivation

Because the reinstatement of the latent task is not required for the current complex task, the reactivation may cause interference in generalizable complex task performance. Indeed, previous studies have suggested that the compositional structure of task representation may benefit flexibility and generalization but may cause interference as one component is shared between different tasks [2,32]. Our study characterizes the temporal dynamics of interference and generalization by showing that latent task reactivation leads to interference effect in early-stage but also leads to gradual improvement in complex task learning in later stages (Fig 3C).

The interference effect (i.e., the significant positive correlation between EEG generalization effect in first block and the residual RT in the first and second blocks, Fig 3C) observed early in the test phase suggests that the reinstatement of the latent task may not be strategic. Rather, it may reflect automatic cued memory retrieval, given that retrieving the associated item can be automatic after practice, such that inhibitory control is needed to suppress the retrieval [53,54]. In this study, the participants encountered each association (i.e., complex tasks) 96 times in the training stage. This level of exposure may facilitate automatic retrieval of the latent task in the test phase.

In contrast to the zero-shot learning findings in previous studies [16–18] showing immediate improvement in performance after successful inference, our analyses showed a gradual improvement of performance in generalizable complex tasks. We propose that this surprising finding could be due to two reasons. First, as we observed the reinstatement of the latent task is linked to both interference in early trials and generalization in late trials of the test phase (Fig 3C), it is possible that the resolution of the interference hinders early manifestation of the generalization effect. Second, unlike previous studies, associative inference in this study applied to task performance, which involves procedural memory [55]. As learning in procedural memory is typically gradual [6,55], we speculate that the reported generalization effect could be decelerated by the involvement of procedural memory. These processes involving resolution of interference and learning of procedural memory may result in slow generalization processes, contrasting to the fast generalization effects that are reported previously [16–18].

## Connections to cognitive control research

The current study bridges a link between memory literature and cognitive control domain. The activation of associated memory items, even when not relevant to the current task, has been extensively reported in the memory literature [8,39,43]. The finding of activation of latent task activation shows that task learning may leverage the same cognitive and neural mechanisms that construct and navigate cognitive maps of abstract spaces to guide behavior. From this viewpoint, task representation can be modeled as a trajectory through a structured, map-like task space, mainly in hippocampal-orbitofrontal cortex network [23,32,56–58]. The cognitive control function has been known to play a critical role in translating information from cognitive maps into actionable production rules via the frontoparietal network [23,32]. The current study extends this finding by showing that cognitive control functions, which heavily involves lateral prefrontal areas, are supported by latent task reactivation through associative memory, which is believed to be supported by the MTL.

Our findings provide evidence for compositionality underlying the flexible learning of complex tasks. Compositionality refers to the ability to separate task representations into reusable components, which can be used as basic building blocks and re-combined to form new task representation. Compositionality has been hypothesized as a core process that enables efficient learning by leveraging existing representations to build higher-level ones [2,3,59,60], which enables flexible and efficient learning of tasks [23,32]. Supporting this claim, previous literature in human studies showed that the prefrontal cortex (PFC) holds high-dimensional task representations that can be separated into multiple combinations according to the current requirements for the task [4,14,32,59,61]. This high-dimensional task representations in PFC enables us to separate and recombine flexibly, which enables to learn new tasks efficiently by generalizing existing task representations.

When understanding the decoding results, a central question is what aspects of a task were decoded. One might interpret this decoded information as simply reflecting the concrete, early-stage processing of concrete stimulus features in simple tasks (e.g., the oval shape), rather than abstract type of representation (e.g., shape processing). While the decoded information likely involves early-stage processing, we argue that this representation is abstract for two main reasons. First, the trained and tested EEG data do not share same response rules (simple task: 'matching' decision, complex task: 'exclusive or' decision), indicating that the decoded information is not specifically involved to a response related process. Second, the decoders were trained at the task level using trials with different stimuli. Thus, the decoded representation was not specific to a specific cue or feature value (e.g., 'red') but rather reflected the cognitive process of guiding feature-based attention along the encompassing feature (e.g., color). Overall, we argue that this representation is 'abstract' because it represents a higher-order rule that encompasses specific feature values and cognitive processes that flexibly guide these feature-based attention.

In conclusion, we provide electrophysiological evidence that the reinstatement of simple tasks supports complex task learning in hierarchical task organization. Specifically, we report the decoding of simple task representations from the complex task EEG data, which suggests that complex task representations include constituent simple task representations. Furthermore, we found significantly above-chance decoding of latent task representation from the EEG data of complex tasks that consisted of indirectly associated simple tasks, which suggests that participants generalized complex task associations via associative inference and the reinstatement of the latent simple task. Further supporting this conclusion, this latent task representation in the complex task EEG data was significantly related to the EEG measure of association effect and behavioral generalization effect. Together, these findings extend our understanding of hierarchical task organization that supports efficient task learning.

## Materials and methods

### Ethics statement

The study was approved by the University of Iowa Institutional Review Board (IRB #: 202001312). All experimental procedures were performed in compliance with the Declaration of Helsinki. Participants provided written consent before participating.

## Participants

Forty-five healthy young adults participated in the experiment and received compensation at a rate of $25/hour. Four participants were excluded from further analysis due to excessive nonstereotypical artifacts in the EEG data. One participant was excluded from further analysis due to lower than 70% accuracy in a specific feature association during every block in training stage. The final sample consisted of 40 participants (23 females and 17 males; age: M = 22.69, SD = 4.35).

## Stimuli

We used compound stimuli composed of six features. Each feature had two possible values (Fig 1A). The features and their values are as follows: shape (oval or rectangle), color (green or red), outline (thick or thin), shadow (cast upward or downward), tilt (clockwise or counterclockwise), and pattern (parallel or diagonal). This design leads to 64 unique stimuli. Other stimuli included word and image cues indicating each of the feature values (Fig 1A).

## Procedure

The experimental procedure was similar to Lee, Hazeltine, and Jiang [6]. The experiment consisted of two phases: a simple task phase (8 blocks of 48 trials each) followed by a complex task phase (11 blocks of 48 trials each). Participants had self-paced resting periods between blocks and had at least 5 min of rest between phases.

Each simple task trial started with the presentation of a cue of a stimulus feature value (e.g., filling pattern is diagonal) for 1,500 ms (Fig 1B). The cue alternated between image and word format (Fig 1A) after each trial so there were no cue repetitions. The cue was followed by the presentation of a fixation cross for 500 ms, after which the target stimulus was presented until response or a response deadline. The initial response deadline was set to 1,500 ms and was adjusted based on a staircase procedure. Specifically, the response deadline decreased by 50 ms when the participants made one or no errors and increased by 50 ms when participants made three or more errors in the last five trials. Otherwise, the response deadline remained unchanged. The participants were instructed to judge whether the cue matched the feature of the target stimulus by pressing the 'D' or 'K' button for yes or no, respectively. After each response, participants received feedback (correct or incorrect, presented in the center of the screen) for 500 ms. From the second block on, the feedback also included overall task accuracy. As a cue indicates which feature is task-relevant, the cues defined six different simple tasks (one for each cue). The simple tasks were coded using letters A-F (Fig 1C), with the simple task-letter mapping randomized for each participant.

The complex task was similar to the simple task except for three modifications (Fig 1B). First, the first screen showed two cues for 2,500 ms. Second, the rule of the complex task followed the 'exclusive or' rule. Specifically, participants were required to press the 'D' button if only one cue matched the target stimulus and the 'K' button otherwise. This rule was employed to ensure that the correct response could only be made after evaluating both cues. Third, the initial response time deadline was set to 2,500 ms for the staircase procedure. A complex task can be viewed as a combination of two simple tasks and was used to investigate association and generalization in hierarchical task representation. Thus, we coded a complex task using constituent simple tasks (e.g., complex task AB is composed of simple tasks A and B).

The complex task phase consisted of two different stages (Fig 1C): a training stage (blocks 1–8) and a test stage (blocks 9–11). To study generalization at the complex task level, the simple tasks were divided into two groups (A-C and D-F). In the training stage, participants performed complex tasks AB and DE in odd runs and complex tasks BC and EF in even runs. In the test stage, participants performed generalizable complex tasks AC and DF, as well as nongeneralizable control complex tasks AD and CF. Complex tasks AC and DF are generalizable because the constituent simple tasks were practiced with a common simple task (e.g., B and E) during the training stage. In contrast, control complex tasks AD and CF cannot be inferred by complex tasks learned in the training stage based on integrative encoding theory [20,22,25] or the recurrent interaction of association theory [21].

## Behavioral analysis

The behavioral analysis focused on replicating the generalization effect in Lee, Hazeltine, and Jiang [6]. Specifically, we tested the generalization effect of faster learning of the generalizable than control complex tasks (Fig 1D) using the analysis from Lee, Hazeltine, and Jiang [6]. We first removed error trials and trials with RTs that were above 5 standard deviations from the subject-mean. Next, we built a nuisance effect design matrix consisting of trial-wise status of response, response repetition, post-error, cue modality (text or image), task repetition, cue modality repetition, and complex task, and regressed out the effects of these confounding factors from RTs in the test stage. Lists of confounding factors and their VIF scores are listed in supplemental material (S2 Table). Next, we fit a design matrix, which consisted of binary regressors marking generalizable and control complex tasks (to estimate RTs at the beginning of test stage) and their temporal order (to estimate RTs change over trials) against the residual RTs. These steps were performed separately for each participant. We compared the individual regression coefficients of estimated RT improvement over trials (slope) between regressors representing generalizable and control conditions of feature associations at the group level using paired $t$-tests.

## Linear mixed-effect model analysis for complex task in training stage

To test the learning effect in the training stage of complex task, we performed a linear mixed-effect model analysis. The analysis was performed on RTs and accuracy separately using R software version 4.1.0 [62] with lme4 package [63]. The formula for this analysis is as follows:

$$y_i = (\beta_0 + u_{i1}) + (\beta_1 + u_{i2})x + \varepsilon_i$$

where $y_i$ refers to a value of behavior measures (either block-mean RT or accuracy) for a participant $i$. $\beta_0$ and $\beta_1$ each represents fixed effects for the intercept ($\beta_0$) and slope ($\beta_1$) that are consistent among participants (i.e., fixed effects). Additionally, $u_{i1}$ and $u_{i2}$ represent random effects for the intercept and slope that can vary for each participant. The term $x$ represents block, and $\varepsilon_i$ reflects errors. In this model, the effect of interest was the fixed-effect of slope, $\beta_1$, that indicates how the behavioral measure changes over blocks.

In the simple task, we found that RTs consistently decreased over blocks ($\beta_1 = -0.04$ (0.002), $t(39) = -20.45$, $p < .001$, Cohen's $d = -3.27$). Accuracy exhibited a significant, yet small decline over time ($\beta_1 = -1.58$ (0.153), $t(39) = -10.33$, $p < .001$, Cohen's $d = -1.65$). The decline in accuracy likely reflects a speed-accuracy tradeoff resulting from the staircase procedure constantly reducing the response deadline (see Materials and methods). Specifically, the staircase procedure will increase the difficulty by reducing the response deadline if the accuracy was less than 80% in the last five trials. Nevertheless, group-average accuracy remained above 80% in all blocks. In the training stage of complex task, we found a pattern identical to the simple task for both RTs (Odd: $\beta_1 = -0.04$ (0.005), $t(39) = -7.53$, $p < .001$, Cohen's $d = -1.21$; Even: $\beta_1 = -0.05$ (0.004), $t(39) = -13.39$, $p < .001$, Cohen's $d = -2.14$) and accuracy (Odd: $\beta_1 = -0.01$ (0.003), $t(39) = -2.48$, $p = .017$, Cohen's $d = -0.40$; Even: $\beta_1 = -0.02$ (0.002), $t(39) = -7.61$, $p < .001$, Cohen's $d = -1.22$) while maintaining a block-wise average accuracy of over 80%. Similar to the simple task phase, the decrease in both RT and accuracy is likely a result of speed-accuracy tradeoff as the response deadline shortened. Overall, the behavioral data indicate that the participants performed the simple and complex tasks as instructed.

## Linear mixed-effect model analysis for complex task in test stage

For the confirmatory analysis of the generalization effect in the test stage of complex tasks, we performed linear mixed-effect model analysis. The formula for this analysis is as follows in R module with 'lme4′ package:

$$RTs \sim 1 + \textit{trial\_type}\ (\textit{gen vs con}) * \textit{trial\_index}\ (\textit{RTs slope}) + \textit{post\_error} + \textit{response\_repetition} + \textit{task\_repetition} + \textit{Cue\_modality} + \textit{Complex\_Task\_Identity} + (1 \mid \textit{Subject})$$

We focused on the interaction between 'trial_type' and 'trial index', which represents the difference of RTs slope over trials between generalizable and control complex tasks. The other covariates were identical to the main behavioral analysis (see S2 Table). We found a significant negative interaction effect between trial_type and trial_index. The result was consistent when we included random effect of RTs slope ($t$(4379.3488) = −2.055, $p$ = .040), but model failed to converge. This result suggests that the participants showed significantly faster reduction in RT over time for generalizable complex tasks than control complex tasks, which is consistent with our main behavioral analysis.

## EEG recording and preprocessing

EEG data were recorded using a 64-channel active Ag/AgCl electrodes connected to an actiCHamp plus amplifier (BrainVision) at a rate of 500 Hz (10 s time-constant high-pass and 1,000 Hz low-pass hardware filters). The reference and ground were set to Pz and Fpz electrodes, respectively. Raw EEG data were preprocessed using customized MATLAB scripts and the EEGLAB toolbox [64]. The EEG data were first band-pass filtered from 0.3 to 50 Hz using Hamming windowed sinc FIR filter (pop_eegfiltnew.m). From the filtered continuous EEG data, we removed epochs with nonstereotypical artifacts for further analysis using outlier statistics and visual inspection. The remaining EEG data were re-referenced to the common average and decomposed using the independent component analysis (ICA) to reject components reflecting eye movements and electrode specific artifacts. After excluding the artifact components, the EEG data were reassembled and epoched relative to the target stimulus onset (simple task: −1,000 to 1,250 ms, complex task: −1,000 to 2,000 ms) for further analysis (Fig 4A). The choice of using target stimulus locked data was to control for the confounds of perceptual information when training and testing the decoders (see below). The preprocessed EEG data were smoothed and down-sampled to 20 Hz to reduce computation time for the decoding analysis and noise in the data.

## EEG decoding analysis

We trained decoders to discriminate among the six simple tasks at each time point using trial-level EEG data from all 64 electrodes (Fig 4B). The decoders were L2 regularized multinomial linear regression with a tolerance (tol) of $1 \times 10^{-4}$ and inverse of the regularization strength of 1 [65]. Decoder training and test were implemented using the scikit-learn package [66] and customized Python scripts. This approach was applied to 5-fold cross-validation using simple task data for each participant. The performance of the decoders was assessed by comparing the logit-transformed probabilities to logit-transformed baseline decoding performance (i.e., 1/6) using a one sample $t$-test.

Following cross-validation, a decoder was trained on each time point using all simple task data (Fig 4C) and tested on each time point and each trial of the complex task data [67–70] to assess the representation strength of simple tasks during the execution of complex tasks (Fig 4D). As a result, for each test, the decoder generated a vector of six classification probabilities, each representing the decoder's belief of a simple task being represented in the complex task at the time point of test. These probabilities were transformed with multinomial logit transformation for further analysis. The decoding results were stored in a four-dimensional tensor for each participant in the format of [trials in the complex task] × [time points of simple task] × [time points of complex task] × [6 simple tasks] for each trial of the complex task phase.

## Extracting pattern matrix from decoding analysis

Using the weight matrix (W) from the decoding analysis, we computed the pattern matrix (A) to quantify electrode involvement in decoding at each time point, ensuring better interpretability and biological relevance [34]. The calculation is given by:

$$A = \Sigma_x W \Sigma_s^{-1}$$

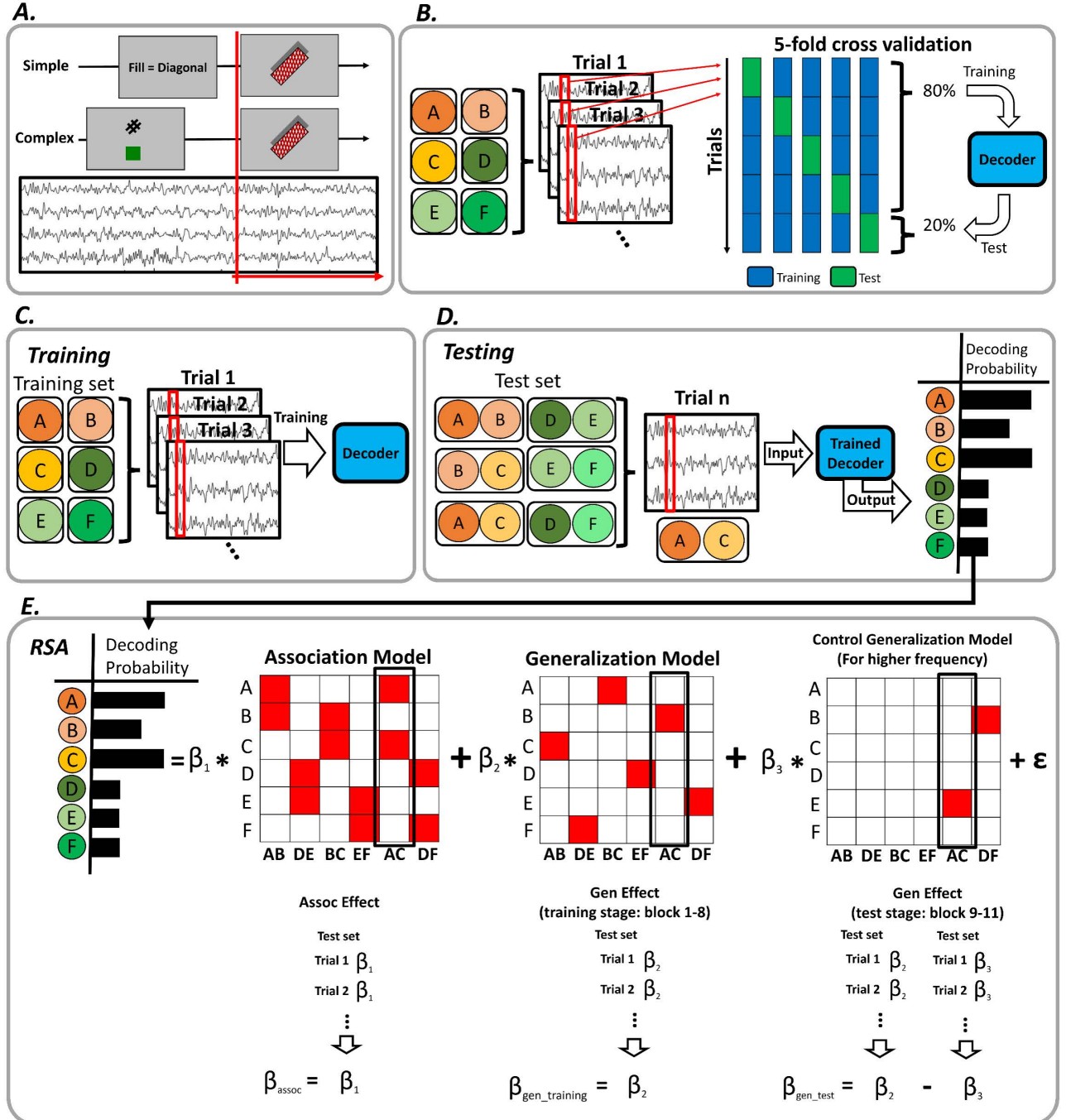

**Fig 4. EEG decoding analysis procedure. (A)** Illustration of the epochs for the EEG data analysis. The EEG epochs were time-locked to the onset of the target (red lines) to de-confound the effect of perceptual information on decoding performance. **(B)** Decoding analysis using simple task EEG data. At each time point, we trained and tested a decoder of six simple tasks using 5-fold cross-validation. **(C)** For cross-phase decoding analysis, the decoder was trained on simple task EEG data. **(D)** The trained decoder was tested using complex task EEG data and generated a decoding probability for each of the six simple tasks. **(E)** The RSA approach. The logit-transformed decoding probability was regressed against two model predictors representing association and generalization effects, and one control regressor representing unassociated latent task in test stage. This control regressor was included to control for higher frequency of latent task in test stage. This procedure was repeated on each complex task trial. We averaged the trial-wise association and generalization effects for each participant and then ran a one sample *t*-test against 0 for the association effect and the generalization effect in training stage. For the generalization effect in test stage, a paired *t*-test was performed against control model. The data and code needed to generate this figure can be found at https://doi.org/10.17605/OSF.IO/MZF4A.

where W is the decoder's raw weight matrix, $\Sigma_x$ is the covariance of the EEG data across electrodes, and $\Sigma_s$ is the covariance of the latent source estimates at each time point. This calculation generates a pattern matrix A that captures latent factors that are projected onto the electrodes. This method allows direct interpretation of the spatial contributions of each electrode to the decoded signal.

## Representational similarity analysis

For the RSA, we first tested association and generalization effects separately using the EEG data (Fig 4E). The association effect is based on the hypothesis that neural representations of the complex task should include the encompassing simple tasks (e.g., AC = A + C). The generalization effect is built under the prediction that neural representations of the generalizable complex task should reinstate the latent task (e.g., simple task B is expected to be activated during complex task AC, if AB and BC are generalized). Each effect is represented as a predictor (i.e., a column in Fig 4E). Additionally, there was a control regressor, which assessed the reinstatement of unassociated latent task (e.g., E in AC complex task). This control regressor was included in the model to control the potential confounding effect of generalization effect in test stage, caused by higher presentation frequency of latent task during training stage. Additionally, a vector assigned with averaged RTs of each simple task was included as nuisance regressor in each trial.

Next, all predictors were included in a linear model, which was regressed against the logit-transformed classification probabilities from each trial of the complex task EEG data. From the output of linear regression, we acquired trial-wise *t*-value for each predictor, which represented the association/generalization effect on the current trial. For each regressor, the *t*-values were grouped as a vector with the length of trials in the complex task. In this vector, trials that showed extreme *t*-values (± 5 SD) were excluded following a previous protocol [10]. Each vector from the association and generalization effects was averaged to examine the overall fitness of the regressor across trials. The regression was conducted on decoding probabilities obtained when the decoder was trained on each time point of the simple task EEG data and tested on each time point of the complex task EEG data. In the end, these analyses generated a t-value matrix with the size of the number of time points of simple task × the number of time points of complex task for each participant. For the association effect and generalization effect in training stage, the statistical significance was calculated by conducting one sample *t* test across participants against 0 at each of combinations of simple task and complex task time points. For the generalization effect in test stage, the statistical significance was calculated by conducting paired *t*-test across participants between generalization effect and control regressor effect at each of combinations of simple task and complex task time points. As a confirmatory post-hoc analysis, EEG generalization in test stage was also calculated without control regressor (Fig 3E).

## Cluster-based permutation test

To control for multiple comparisons in the above the final *t*-value matrix, we conducted a cluster-based permutation test with customized python script based on Maris and Oostenveld [35]. Specifically, with simple task EEG data, we repeated a 5-fold cross-validation decoding analysis using shuffled labels of the simple tasks 10,000 times. We pooled the clusters with the largest sum of absolute *t*-values (t-sum) from each repetition to create a null distribution of t-sums, which takes into consideration of both the cluster size and statistical significance. From the actual result, the cluster with the largest t-sum is compared to the null distribution to obtain corrected *p*-value.

For the RSA with one sample *t*-test (e.g., B in AC), we repeated the analysis using shuffled labels of the simple tasks before training the decoder 10,000 times while keeping the labels of the complex tasks the same between repetitions. The remaining procedure was identical to the nonparametric permutation test for simple task decoding analysis.

For the RSA with paired *t* test (e.g., B in AC − E in AC), we first randomly chose 20 among 40 participants, and swapped the labels of two conditions and performed paired *t*-test. From the result, we pooled the cluster with the largest absolute t-sum and created null distribution by repeating 10,000 times. We calculated corrected *p*-value by comparing the actual result from the null distribution.

## Supporting information

**S1 Fig. Behavioral results from the simple task phase.** Individual RT (left) and accuracy (right), imposed with the group mean and SEM, are plotted as a function of block. Participants showed faster RTs over blocks on correct trials, while the accuracy declined after block 2. The decline in accuracy may be attributed to the decrement of the response deadline due to the staircase procedure of adjusting response deadline. Data underlying this figure can be found in the OSF repository (https://doi.org/10.17605/OSF.IO/MZF4A).
(TIF)

**S2 Fig. Behavioral results from the training stage of complex task phase.** Individual RT (left) and accuracy (right), imposed with the group mean and SEM, are plotted as a function of block. Participants showed faster RTs over blocks on correct trials, while the accuracy declined after block 3. The decline in accuracy may be attributed to the earlier response deadline due to the staircase procedure of adjusting response deadline. Data underlying this figure can be found in the OSF repository (https://doi.org/10.17605/OSF.IO/MZF4A).
(TIF)

**S3 Fig. Simple task EEG data decoding results for each task.** Blue flat line indicates chance-level decoding accuracy. Data underlying this figure can be found in the OSF repository (https://doi.org/10.17605/OSF.IO/MZF4A).
(TIF)

**S4 Fig. Simple task EEG data temporal generalization decoding results for each simple task.** Black contour highlights clusters that showed significant time points after FDR correction. Data underlying this figure can be found in the OSF repository (https://doi.org/10.17605/OSF.IO/MZF4A).
(TIF)

**S5 Fig. Frequency of tasks assigned to specific labels across subjects.** Data underlying this figure can be found in the OSF repository (https://doi.org/10.17605/OSF.IO/MZF4A).
(TIF)

**S6 Fig. Result of decoding of nonassociated latent task representations in test stage (e.g., E in AC).** The result shows positive values between 1,000 and 2,000 ms in complex task EEG data, which partially overlaps with post-hoc EEG generalization analysis (Fig 2D). However, the largest cluster in the time window did not survive permutation test ($p = .120$). This result provides preliminary evidence that latent task reactivation after 1,000 ms in complex task EEG data might be confounded by the effect caused by higher presentation frequency of latent task (e.g., simple task B presented twice often compared to A or C during training stage). Data underlying this figure can be found in the OSF repository (https://doi.org/10.17605/OSF.IO/MZF4A).
(TIF)

**S1 Table. Behavioral performance of each simple task in the simple task phase.** Values are reported as the 'mean (standard error of mean)'. Data underlying this figure can be found in the OSF repository (https://doi.org/10.17605/OSF.IO/MZF4A).
(DOCX)

**S2 Table. VIF scores of behavioral confounding regressors.** Values are reported as the 'mean (standard deviation)'. Data underlying this figure can be found in the OSF repository (https://doi.org/10.17605/OSF.IO/MZF4A).
(DOCX)

**S3 Table. Raw reaction times of test stage by block (in ms).** Values are reported as the 'mean (standard error of mean)'. Data underlying this figure can be found in the OSF repository (https://doi.org/10.17605/OSF.IO/MZF4A).
(DOCX)

## Acknowledgments

We thank the members of the Cognitive Control Collaborative and the members of the Human Perception and Performance Group at the University of Iowa for helpful discussions.

## Author contributions

**Conceptualization:** Woo-Tek Lee, Eliot Hazeltine, Jiefeng Jiang.

**Funding acquisition:** Jiefeng Jiang.

**Investigation:** Woo-Tek Lee, Jiefeng Jiang.

**Methodology:** Woo-Tek Lee, Jiefeng Jiang.

**Supervision:** Eliot Hazeltine, Jiefeng Jiang.

**Visualization:** Woo-Tek Lee.

**Writing – original draft:** Woo-Tek Lee.

**Writing – review & editing:** Woo-Tek Lee, Eliot Hazeltine, Jiefeng Jiang.

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
