## [Editor Report · Decision Letter 0]

10 Nov 2025

Dear Dr Jiang,

Thank you for submitting your manuscript entitled "Decoding task representations that support generalization in hierarchical task." for consideration as a Research Article by PLOS Biology.

Your manuscript has now been evaluated by the PLOS Biology editorial staff as well as by an academic editor with relevant expertise and I am writing to let you know that we would like to send your revised manuscript submission back to the reviewers.

Once your full submission is complete, your paper will undergo a series of checks in preparation for peer review. After your manuscript has passed the checks it will be sent out for review. To provide the metadata for your submission, please Login to Editorial Manager (https://www.editorialmanager.com/pbiology) within two working days, i.e. by Nov 12 2025 11:59PM.

Kind regards,

Christian

Christian Schnell, PhD

Senior Editor

PLOS Biology

cschnell@plos.org

---

## [Decision Letter · Decision Letter 1]

17 Dec 2025

Dear Dr Jiang,

Thank you for your patience while we considered your revised manuscript "Decoding task representations that support generalization in hierarchical task." for publication as a Research Article at PLOS Biology. This revised version of your manuscript has been evaluated by the PLOS Biology editors, the Academic Editor and two of the original reviewers.

Based on the reviews and on our Academic Editor's assessment of your revision, we are likely to accept this manuscript for publication, provided you satisfactorily address the following data and other policy-related requests:

* We would like to suggest a different title to improve its accessibility for our broad audience:

Neural traces of composite tasks in complex task representation in the human brain reflects learning performance

* Please include the approval/license number of the ethical approval for the experiments.

* Please include information in the Methods section whether the study has been conducted according to the principles expressed in the Declaration of Helsinki.

* Please specify whether the participants provided written or oral consent.

* DATA POLICY:

Regardless of the method selected, please ensure that you provide the individual numerical values that underlie the summary data displayed in the following figure panels as they are essential for readers to assess your analysis and to reproduce it: 1E, 3A, S1 and S2.

* CODE POLICY

* Please move references in the supplementary information to the main reference list. Otherwise these citations won't be picked up by the literature databases.

* Please move the methodological details from the supplementary information to the main manuscript file as well. We do not have a word count limit.

We expect to receive your revised manuscript within two weeks.

*Published Peer Review History*

*Press*

Sincerely,

Christian

Christian Schnell, PhD

Senior Editor

cschnell@plos.org

PLOS Biology

Reviewer remarks:

Reviewer #1: The authors extensively revised their paper in response to the points I raised, and carefully considered all of my suggestions. They included multiple additional analyses and revised their framing in light of the new results. In my view, this has substantially strengthened this work. I am happy to recommend publication.

Reviewer #2: The authors have provided excellent responses to my comments. I am happy to recommend the publication of this manuscript!

---

## [Editor Report · Decision Letter 2]

7 Jan 2026

Dear Dr Jiang,

Thank you for the submission of your revised Research Article "Neural traces of composite tasks in complex task representation in the human brain reflects learning performance" for publication in PLOS Biology. On behalf of my colleagues and the Academic Editor, Huan Luo, I am pleased to say that we can in principle accept your manuscript for publication, provided you address any remaining formatting and reporting issues. These will be detailed in an email you should receive within 2-3 business days from our colleagues in the journal operations team; no action is required from you until then. Please note that we will not be able to formally accept your manuscript and schedule it for publication until you have completed any requested changes.

While you attend to these requests, please also ensure that the figure legends in your manuscript include information on where the underlying data can be found. For example: "The data and code needed to generate this figure can be found at "https://doi.org/10.17605/OSF.IO/MZF4A."

PRESS

Sincerely,

Christian

Christian Schnell, PhD

Senior Editor

PLOS Biology

cschnell@plos.org